# Assignment of a Reference Value of Total Cow’s Milk Protein Content in Baked Cookies Used in an Interlaboratory Comparison

**DOI:** 10.3390/foods11060869

**Published:** 2022-03-18

**Authors:** Andreas Breidbach, Jørgen Vinther Nørgaard, Elena Cubero-Leon, Maria Jose Martinez Esteso

**Affiliations:** 1European Commission, Joint Research Centre (JRC), 2440 Geel, Belgium; jorgen.norgaard@ec.europa.eu (J.V.N.); elena.cubero-leon@ec.europa.eu (E.C.-L.); mjmme20@gmail.com (M.J.M.E.); 2Department of Agrochemistry and Biochemistry, University of Alicante, Carrera de San Vincente del Raspeig s/n, San Vincente del Raspeig, 03690 Alicante, Spain

**Keywords:** reference value, interlaboratory comparison, milk protein, traceability, measurement uncertainty, LC-IDMS

## Abstract

Interlaboratory comparisons (ILC) in the food allergens field mainly rely on the use of consensus values per applied methodology or even per type of an ELISA test kit. Results suggest good reproducibility; however, possible biases may not be recognized since metrological traceability to an independent reference is lacking. The work presented here utilizes isotope dilution mass spectrometry (IDMS) to assign a reference value of the total cow’s milk protein (TCMP) content in a baked cookie and its associated uncertainty. TCMP consists of several individual proteins, of which five (representing 92%) served us as markers for TCMP. Per marker, one to four proteotypic peptides were selected for the quantification. These were synthesized, and the mass fractions of respective reference solutions were determined with peptide-impurity-corrected amino acid analysis to establish traceability to SI units. Stable isotope labelled (“heavy”) analogues of the proteotypic peptides were also synthesized and blended with extracts of the test material or the reference solutions for IDMS. Through careful measurement design minimizing biases, well-defined model equations were developed, allowing appropriate estimation of the associated uncertainty. The determined reference value of 11.8 ± 1.1 mg TCMP/kg cookie was used for scoring of a novel ILC.

## 1. Introduction

The unintentional presence of certain allergenic foodstuffs in the food consumed by sensitive individuals could be a major threat to their health. Many food business operators use precautionary allergen labelling (PAL) to alert consumers of the potential presence of allergenic proteins in the food. Since there is no cure for patients suffering from such food allergies, the only viable countermeasure is the complete avoidance of this food. In order to avoid unnecessary use of PAL statements, which would limit the food choice of allergic individuals, quantitative risk assessments [1] using well-established reference doses are recommended [2]. In this context, reliable measurement data are needed to determine the levels of food allergens present in food.

Since such measurements will most likely be performed by different laboratories across the EU and the world, it is of utmost importance that the testing laboratories can demonstrate their competence. This includes providing accurate and reliable measurement results on the content of food allergens in the decision-relevant reporting quantity (i.e., the relevant measurand). Laboratories can make use of inter laboratory comparisons (ILC), in particular proficiency tests, to demonstrate such capabilities [3].

Currently ILC providers in the food allergen arena commonly use a consensus value for scoring, which is often applied for each methodology, such as individual ELISA test kits [4]. This only allows for the comparison of results obtained applying the same or a similar analytical method. In order to assess results derived from different methodologies, the reporting in form of a common measurand is necessary, further supported by an independent reference value.

In 2017, the European Network of Food Allergen Detection Laboratories (ENFADL) was inaugurated with the aim of harmonizing EU food allergen measurements. Food control laboratories from most of the EU Member States are participating in this network. One of the first network activities was the organization of an ILC. This was a major step towards achieving comparable measurement results. Regardless of the testing methodology used by an ILC participant, the results had to be reported in the common measurand, which was mg of total cow’s milk protein (TCMP) per kg of a baked cookie [5].

To determine such a reference value, one needs a reference method of analysis. In clinical chemistry, liquid chromatography-mass spectrometry (LC-MS) has become the “gold standard” [6,7]. For the determination of proteins and peptides, LC-MS can overcome the limitations of immunoassays, but its application to developing reference methods for protein analysis is only now becoming more prominent [8]. MS has the unique capability to distinguish peptides and their isotopologues (analogues that are enriched/depleted in one or more elemental isotopes), which otherwise behave identical during sample preparation. Adding such isotopologues in known amounts to test materials and measure them with MS is known as isotope dilution mass spectrometry (IDMS) [9,10]. IDMS, if executed properly, is a primary ratio method that provides the shortest traceability to SI units and, as such, reference values with the lowest uncertainties [11,12].

The aim of this paper is to describe in detail and discuss the assignment of the reference value (*x_a_*) with LC-IDMS [13] for the test material used in this ILC. Furthermore, a full uncertainty budget is provided for the estimation of the measurement uncertainty associated with this reference value.

## 2. Materials and Methods

### 2.1. Materials

Ammonium bicarbonate (AmBic), urea, dithiothreitol (DTT), iodoacetamide (IAA), dimethylsulfoxide (DMSO), and formic acid (FA) were obtained from Sigma-Aldrich (www.sigmaaldrich.com, accessed on 16 March 2022). Trypsin (Gold MS grade) was purchased from Promega (Madison, WI, USA). LC-MS grade acetonitrile (ACN) was obtained from VWR (www.vwr.com, accessed on 16 March 2022). Thermo Scientific HyperSep™ 200 mg C18 SPE cartridges were obtained from Biopolymers (Ulm, Germany). Ultrapure water from a Milli-Q system equipped with a LC-polish cartridge (www.merckmillipore.com, accessed on 16 March 2022) was used to prepare the mobile phase. Eppendorf 2.0 mL LoBind centrifuge tubes, 50 mL polypropylene conical screw-cap centrifuge tubes, and all other chemicals were also purchased from VWR. If not mentioned otherwise, all buffer solution were prepared in water.

Eleven proteotypic peptides (representative of five major constituents of TCMP) were synthesized by ThermoFisher Scientific (Custom Peptide Service, Merelbeke, Belgium, www.thermofisher.com, accessed on 16 March 2022) (Table 1). Throughout this paper, the 11 peptides will be referred to by the first three letters of their amino acid sequence.

“Heavy” peptides (stable isotopologues of the 11 proteotypic peptides) with either the C-terminal lysine replaced by ^13^C_6_-^15^N_2_-lysine (K*) or the C-terminal arginine by ^13^C_6_-^15^N_4_-arginine (R*) were obtained as crude lyophilizates from JPT Peptide Technologies GmbH (Berlin, Germany). Skimmed milk powder (SMP; 09G010) was purchased from BIOSERVICE Zach GmbH (Gmünd, Austria)

### 2.2. Preparation of Reference Solutions

Stock solutions of the peptides were prepared at nominal 1 mg/g in ACN/0.1 mol/L AmBic (20/80, *v*/*v*), except for ALN, which was prepared in ultrapure water. Stock solutions were stored at −20 °C, and their stability was checked over six months. The peptide purities and sequences were confirmed by analyzing the individual peptide stock solutions by mass spectrometry (Synapt G2 HDMS (Waters, Manchester, UK)) and HPLC-UV. Although the synthesized peptides were high in peptide purity, the lyophilized product contained other chemical impurities such as H_2_O and salts. Therefore, very importantly, the accurate peptide mass fraction of each stock solution was determined by amino acid analysis (AAA) [14] for all peptides but YLG. Of the six amino acids Ala, Ile, Leu, Phe, Pro, and Val, which are amenable to AAA, YLG contains only Leu. That was considered as insufficient for a reliable determination. Since YLG was synthesized in parallel with the other peptides, its purity was estimated as the average of the purities of all the other peptides. The uncertainty of this purity estimate was taken as a rectangular distribution with a width equal to the range of the purities. The mass fraction of YLG was calculated from the gravimetric value corrected by the purity estimate. The associated uncertainty was propagated from the weighing and purity uncertainty estimates.

From the stock solutions, two mixed reference solutions (high and low mass fraction) were prepared. In addition, one mixed spike solution was prepared from the “heavy” peptides. For the “heavy” peptides mix, an accurate knowledge of the mass fractions is irrelevant for double isotope dilution experiments, as long as the same solution is used throughout the whole measurement campaign. All preparations were done gravimetrically with a calibrated analytical balance (with a readability of d = 0.0001 g). In order to reduce adsorption of the reference peptides on the container walls in diluted solutions (in the low nmol/L range), the solutions were prepared in LoBind centrifuge tubes following the suggestions in Hoofnagle et.al. [15].

### 2.3. Preparation of the Test Material

The test material for this ILC was an incurred cookie. It was prepared by mixing dairy-free fat, sugar, flour, sodium bicarbonate, ammonium bicarbonate, and salt to a dough. Skimmed milk powder (10.8 mg TCMP/kg) was added to the wet dough. This value was based on a total nitrogen determination by Dumas. More detailed information about the test material can be found elsewhere [5].

### 2.4. Preparation of Sample and Calibration Blends

The preparation of the blends followed the preparation procedure reported in [13]. A short description is outlined here. For sample blends (SB), 1 g of test material was extracted with 15 g of 5 mol/L urea and 50 mmol/L DTT in 50 mmol/L AmBic pH 8 for 3 h at 4 °C with agitation in 50 mL centrifuge tubes. After spinning down particulate matter, 250 mg of the supernatant was transferred to a 2 mL LoBind centrifuge tube. The extract aliquot underwent chemical reduction for 45 min at 37 °C through the addition of 50 µL 80 mmol/L DTT in 25 mmol/L Ambic. The alkylation reaction was achieved through addition of 50 µL 450 mmol/L IAA in 25 mmol/L AmBic and incubation for 45 min at room temperature in the dark. Finally, 250 mg of the spike solution and 250 µL 25 mmol/L AmBic were added. This solution was further diluted with 690 µL DMSO/25 mmol/L AmBic (21/79, *v*/*v*) and then digested for 14 h at 37 °C with 1 part of trypsin per 50 parts of protein. The enzymatic digestion was stopped by adding 5 µL FA to the vial. All mass determinations were done with the above-mentioned calibrated analytical balance.

The digest was cleaned-up using SPE cartridges (200 mg HyperSep™ C18 columns, Thermo Scientific, Biopolymers, Ulm, Germany), and the eluate was reduced in volume to between 0.1 and 0.5 mL in a vacuum centrifuge. If necessary, the residue was made up to 0.5 mL with ultrapure MS grade water. This final solution was injected into the LC-MS. A total of six SBs were prepared as duplicates from three ILC test sample units selected randomly.

For the two calibration blends (CB), a TCMP-free material was extracted. Prior to the digestion, 250 mg of mixed reference solution was added in place of the 250 µL 25 mmol/L AmBic. The mass fractions of the peptides in the mixed reference solutions were chosen to bracket the expected mass fractions in the test material.

### 2.5. Measurements

The measurements were carried out on an Orbitrap Elite MS (Thermo Scientific, Merelbeke, Belgium) with a heated electrospray ionization source (HESI) connected to a Nexera LC (Shimadzu, Brussel, Belgium). Separation was afforded by an analytical column Excel 1.7 C18 100 × 2.1 mm, 1.7 µm particle size (ACE, Warrington, UK) with mobile phase A: water/FA (999/1, *v*/*v*) and mobile phase B: ACN/FA (999/1, *v*/*v*); a flow rate of 300 µL/min; and a column temperature of 60 °C. The mobile phase gradient started with 8% B and increased linearly to 32% B in 25 min. To prevent fouling up of the column with matrix components, each analytical run was preceded by a pre-run with a DMSO/ACN (80/20, *v*/*v*) injection and a fast gradient from 95–8% B and succeeded by a post-run with another DMSO/ACN injection and a fast gradient from 32–95% B. This set-up (pre-, analytical-, and post-injection) had a total cycle time of 38 min and ensured relative retention time standard deviations of below 1% over the measurement campaign.

Acquisition was done in a scheduled positive selected ion monitoring (SIM) mode where the SIM range equaled the average of the monoisotopic m/z of native and labelled analyte ± m/z 7.5 and an acquisition time window of the expected retention time of an analyte ± 1 min. The resolving power was set to 30,000. The ion source settings were as follows: ion transfer capillary temperature 240 °C, source heater temperature 250 °C, sheath gas flow 30 au (arbitrary unit), auxiliary gas flow 5 au, sweep gas flow 1 au, and source voltage 4 kV. The AGC target was set to 1e5 with a maximum injection time of 200 ms at 1 microscan.

The measurement campaign was designed so that each sample blend (SB) was bracketed by the two calibration blends (CB1, CB2) in the following order: CB1, SB, CB2, SB, CB1, SB, CB2, SB, CB1, SB, and CB2. Six of these blocks for each of the six SBs were run, with a blank injection solution in between, for a total of 71 runs. For each run, the ratios of the peak areas of targeted peptides divided by the peak areas of their respective isotopologues were calculated. The isotope ratios obtained for an SB run were divided by the absolute difference of the isotope ratios obtained from the preceding and succeeding CB runs to obtain the ratio of ratios R’ (see Equation (1)). Finally, the five calculated R’ values per SB were averaged and used for the computation of the peptide amounts. This process is minimizing instrumental influences and is providing smallest possible uncertainties [16,17].

An alternative measurement platform was used for confirmation. The same injection solutions were measured on an LC-MS system consisting of a nanoAcquity LC and a Xevo TQ-S triple quadrupole mass spectrometer (Waters, Manchester, UK). Separation was afforded by a 150 µm x 100 mm Peptide BEH C18 130 Å, 1.7 µm UPLC iKey column with an integrated ESI emitter, which formed part of an IonKey based separation system (Waters, Manchester, UK). Measurements were done in the MRM mode with a dwell-time set to 40 ms and optimized collision energies for all peptides. The sample injection volume was 2 µL using a full loop injection. At a flow rate of 2 µL/min the following gradient was used: 0–5% B for 0.5 min, 5–25% B during 17.5 min, and 25–40% B during 4.5 min. Solvent A was water/FA (999/1, *v*/*v*), and solvent B was ACN/FA (999/1, *v*/*v*). Each sample run was followed by an injection of 2 µL of ACN applying a gradient of 0–100% B for 3 min, 100–0% B for 1 min, and a hold for 4 min, with a flow rate of 10 μL/min, in order to condition and equilibrate the analytical column between sample injections. Comparable results between the two platforms were obtained for the assigned value and the measurement uncertainty (data not shown). This shows that the method is robust and can be transferred to other measurement platforms.

The investigation of extraction and digestion conditions, as well as determining which factors are affecting them, has been reported elsewhere [18]. These factors were optimized for the tested material. The maximum amount of protein was extracted, and completion of the digestion was ensured.

### 2.6. Computations and Statistical Analysis

All computations were performed utilizing the software R, V3.5.1 [19] on a Windows 7 operated desktop PC. A script containing the functions to compute the assigned value and its associated uncertainties was developed in-house and is available upon request.

### 2.7. Assigned Value (x_a_)

Given the knowledge of the *R*′ values, the amount *n_i,j,k_* of the *i*-th peptide of the *j*-th protein in the test solution of the *k*-th SB could be calculated as:(1)$$n_{i,j,k}=m_{Y,k}\;\left(\frac{b_{Zc2,i,j}m_{Zc2}}{m_{Yc2}}-\frac{b_{Zc1,i,j}m_{Zc1}}{m_{Yc1}}\right)\;{\bar{R}}_{i,j,k}^{\prime }.\;$$
with

*m_Y,k_*—mass of spike solution added to *k*-th SB,

*b_Zc_*_2*,i,j*_—molality of the *i*-th peptide of the *j*-th protein in the mixed reference solution 2,

*m_Zc_*_2_—mass of mixed reference solution added to CB 2,

*m_Yc_*_2_—mass of spike solution added to CB 2,

*b_Zc_*_1*,i,j*_—molality of the *i*-th peptide of the *j*-th protein in the mixed reference solution 1,

*m_Zc_*_1_—mass of mixed reference solution added to CB 1,

*m_Yc_*_1_—mass of spike solution added to CB 1,

${\bar{R}}_{i,j,k}^{\prime }$—mean of ratios of the isotope ratio of the *i*-th peptide of the *j*-th protein in the *k*-th SB over the absolute difference of isotope ratios of the respective peptide in CBs 2 and 1 $\left(R^{\prime }=R_{B}/\left|R_{Bc2}-R_{Bc1}\right|\right)$.

Equation (1) applies when the same spike is added to the SBs and CBs.

Each of the proteotypic peptides is present at 1 mol per 1 mol marker protein. A complete digestion is then evidenced by equimolarity of all respective peptides per marker protein. Therefore, the average amount of all measured proteotypic peptides per marker protein was used to calculate its mass fraction in an SB as:(2)$$w_{j,k}=\frac{\sum_{i=1}^{N}n_{i,j,k}}{N_{j}}\frac{m_{Exsolv,k}}{m_{X,k}m_{Extr,k}}M_{j}\;f_{P-P,j,k}.\;$$
with

*w_j,k_*—mass fraction of the *j*-th marker protein in the *k*-th SB,

*n_i,j,k_*—amount of the *i*-th proteotypic peptide of the *j*-th marker protein in the *k*-th SB,

*N_j_*—number of proteotypic peptides of the *j*-th marker protein,

*m_Exsolv,k_*—mass of the extraction solvent added to the *k*-th SB,

*m_X,k_*—mass of the test portion in the *k*-th SB,

*m_Extr,k_*—mass of the extract aliquot used for the *k*-th SB,

*M_j_*—average molecular weight of the *j*-th marker protein (accounting for natural variation of the amino acid sequence),

*f_P-P,j,k_*—unity factor carrying the between-peptide uncertainty (accounting for digestion effects).

Once the mass fractions of the marker proteins were known, the mass fraction of TCMP in a SB could be calculated as:(3)$$w_{TCMP,k}=\overset{5}{\underset{j=1}{\sum }}\left(w_{j,k}\right)\;f_{\varepsilon }$$
with

*w_TCMP,k_*—mass fraction of TCMP in the *k*-th SB,

*f_ε_*—compositional factor accounting for the fractional coverage of the included proteins relative to TCMP.

The assigned value (*x_a_*) is then calculated as the average of the six SBs:(4)$$x_{a}=\frac{\sum_{k=1}^{6}w_{TCMP,k}}{6}\;f_{SB-SB}.\;$$
with

*w_TCMP,k_*—mass fraction of TCMP in the *k*-th sample blend,

*f_SB-SB_*—unity factor carrying the between-SB uncertainty.

### 2.8. Associated Measurement Uncertainty of the Assigned Value (u(x_a_))

In compliance with ISO 13528:2015 [20], the standard uncertainty (*u(x_a_*)) associated with the assigned value was calculated combining the standard uncertainties of the value assignment (*u_char_*) with the ones due to homogeneity (*u_hom_*) and stability (*u_st_*):
(5)$$u^{2}\left(x_{a}\right)=u_{char}^{2}+u_{hom}^{2}+u_{st}^{2}$$

The combined standard uncertainty *u_char_* was estimated by propagating the uncertainties of the four steps “amount of proteotypic peptide in test solution” (*n_i,j,k_*, Equation (1)); the “mass fraction of the marker protein in a SB” (*w_j,k_*, Equation (2)); the “mass fraction of total cow’s milk protein” (*w_TCMP,k_*, Equation (3)); and, finally, the average of the six SBs (*x_a_*, Equation (4)). Since the prerequisites of linearity and/or small relative individual uncertainties were met, the Kragten spread sheet approach [21] was used to calculate the standard uncertainties of *n_i,j,k_*; *w_j,k_*; *w_TCMP,k_*; and *x_a_*.

The uncertainty of the homogeneity study (*u_hom_*) was calculated as the square root of the between-test unit variance obtained from single-factor ANOVA of the ELISA results. Since the stability study confirmed that the test item was stable, the uncertainty due to stability (*u_st_*) was set to zero.

## 3. Results and Discussion

Six sample blends were prepared as described above, and the released peptides were measured with LC-IDMS to determine R′ (see Equation (1)). In the measurement batch, the SBs were bracketed between the CBs to minimize influences caused by changing instrument conditions, such as temperature changes and degradation of the ion source. With these influences minimized or even eliminated, this repetition reduced the associated uncertainty. The relative uncertainties of the R’ values were between 0.2% and 8.3%, with a median of 1.4%. Exceptions were the relative uncertainties registered for NAV of CASA2, which were between 4% and 23%, with a median of 14%. There were no obvious reasons identified for this deviation such as aberrant peak shapes or insufficient signal.

The individual contributions to the respective combined uncertainties are shown in Table 2. For Equation (1), the uncertainties of the mass measurements are negligible. The uncertainty of R′ was minimized through properly designing the sequence of measurements as described above, which leaves the uncertainties of the molalities of the reference solutions as the largest contributor. This shows that an appropriate knowledge of the purity and mass fraction of the reference peptide solutions is of utmost importance to guarantee accurate results. A purity correction (determined by amino acid analysis) was applied to the relevant peptide to determine the mass fractions for 10 of the 11 reference peptide solutions. The relative uncertainties of the mass fractions ranged from 1.1 to 2.5%. For YLG, the estimated relative uncertainty was 16%.

Ideally, all peptides belonging to one protein should be released at the same amount during digestion. In reality, many factors influence the amount of peptide, which can be measured. One of those factors is digestion efficiency, which is not identical for all the individual peptides [18]. Another factor is the stability of the peptides in the digestion solution. Therefore, it is advantageous to measure more than one peptide per marker protein to get an indication of these effects. Figure 1 depicts the calculated amounts of the two measured peptides FFV and YLG of the protein CASA1 with their associated expanded combined uncertainties. The values differ, which is an indication of the effects mentioned above. This difference is not significant because the uncertainty ranges overlap due to the large uncertainty attached to the mass fraction of YLG. Since neither one value can be considered as the “truer” one, the mean of the two is used in Equation (2) (depicted as horizontal bar for SB U167.2 in Figure 1). The relative standard error of this mean is therefore used as the uncertainty associated with the term *f_P-P,j,k_* of Equation (2), to account for these effects. The bold vertical bar in Figure 1 depicts this. This uncertainty is the largest contributor to the combined uncertainty.

For the other four proteins (CASA2, CASB, CASK, and LACB), Figure 2 depicts the calculated amounts and associated uncertainties of the respective peptides. While an overlap of the uncertainty ranges is observed for all CASA2 and some CASB preparations, no overlap is evidenced in the case of LACB. Additionally, for these the uncertainty associated with the term *f_P-P,j,k_* is the largest contributor to the combined uncertainties of the respective mass fractions. For CASK, only one peptide was measured. This leads to uncertainties of the mass fraction, which are smaller than for the other proteins. Smaller uncertainties imply a higher confidence in the result, which is not the case here because only one peptide was measured. Therefore, as a conservative measure, the CASK calculations were penalized with an uncertainty value for *f_P-P,j,k_* of 0.1, which is the average of the uncertainty values of the other caseins. This is depicted as “Penalty” in panel “CASK” of Figure 2.

The marker proteins used in this work were five of the six major constituent proteins of cow’s milk. The missing major constituent is α-lactalbumin (LALBA_BOVIN). Other minor constituents not considered include immunoglobulins and serum albumin. According to Gellrich et al. [23], the five marker proteins cover 92% of the total mass fraction of TCMP. Hence, the compositional factor *f_ε_* used in Equation (3) is equal to 100/92 = 1.087. In order to take into account the unconsidered proteins, a conservative 8% was set as one-half of a rectangular uncertainty range. This implies that there is an equal probability for the coverage to be between 84 and 100%, with 92% being the center. Therefore, the standard uncertainty of the compositional factor *f_ε_* was set to 8/(92 × √(3)) = 0.0502. The TCMP mass fractions calculated for the six SB preparations are listed in Table 2 and depicted in Figure 3. Averaging the six results led to the assigned value.

The standard uncertainty due to the value assignment (*u_char_*) was calculated as 0.473 mg/kg. Since two different ELISA kits were used for the homogeneity study, the largest between-test unit variances were used for *u_hom_* (0.283 mg/kg). By applying Equation (5), the associated combined standard uncertainty was calculated as
(6)$$u\left(x_{a}\right)=\sqrt{0.473^{2}+0.283^{2}+0^{2}}=0.551\;\mathrm{mg}/\mathrm{kg}$$

This results in an assigned value and associated expanded measurement uncertainty equal to:

11.8 ± 1.1 mg of total cow’s milk protein per kg of baked cookie (with expansion factor k = 2).

Knowing the TCMP content added into the dough (10.8 mg/kg) and the moisture loss during baking (16%), a nominal value of 12.9 mg/kg in the baked cookie is estimated. This value is in agreement with the assigned value (*x_a_*), within the measurement uncertainties, which further validates our estimated uncertainty.

## 4. Conclusions

The use of a PAL statement is aimed at ensuring that food placed on the market within the EU is safe for allergic consumers. However, it is recommended these statements are used after proper quantitative risk assessment to avoid further limiting their food choices. This requires reliable results associated with realistic measurement uncertainty estimates. The reference LC-MS method described by Martinez et al. [13] was used to determine an independent reference value (*x_a_*) of mg of total cow’s milk protein (TCMP) per kg of a baked cookie. The detailed knowledge of this analytical method allowed the compilation of the necessary model equations and the respective input quantities (parameters). The uncertainty propagation law applied according to the GUM (ISO 98) [24] allowed for the identification of the major contributors to the combined uncertainty, which were the molality of the individual peptides used in the reference mixture solution (*b_Zc_*_2*,*1*,*1_), and the factor for the missing 8% total milk protein. This uncertainty budget enabled a rigorous estimation of the measurement uncertainty (of 4.7%) and supports the metrological traceability of the assigned reference value.

The estimation of an assigned range can be used in an assessment of the performance of laboratories analyzing the test item regardless of the methodology applied. Furthermore, laboratories that perform well can, therefore, provide reliable measurement results needed to perform correct quantitative risk assessments to the benefit of the allergic consumers. The approach described in this manuscript could be adapted to other quantitative methods developed for food allergen analysis.

## Figures and Tables

**Figure 1 foods-11-00869-f001:**
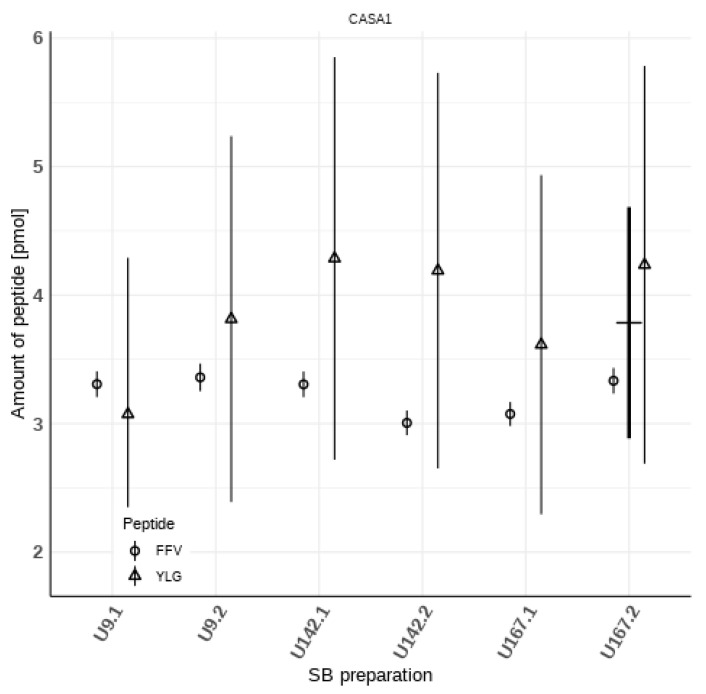
Measured amounts of peptides FFV (i = 1) and YLG (i = 2) of protein CASA1 (j = 1) in the six SBs (index k = 1 to 6, left to right); symbols depict the quantity values of the respective peptides, and the vertical bars the associated expanded uncertainties (expansion factor k = 2); as an example, the horizontal and bold vertical bar for SB U167.2 depict the mean of the two quantity values and the absolute expanded uncertainty of *f_P-P,_*_1*,*6_ (Table 2), respectively.

**Figure 2 foods-11-00869-f002:**
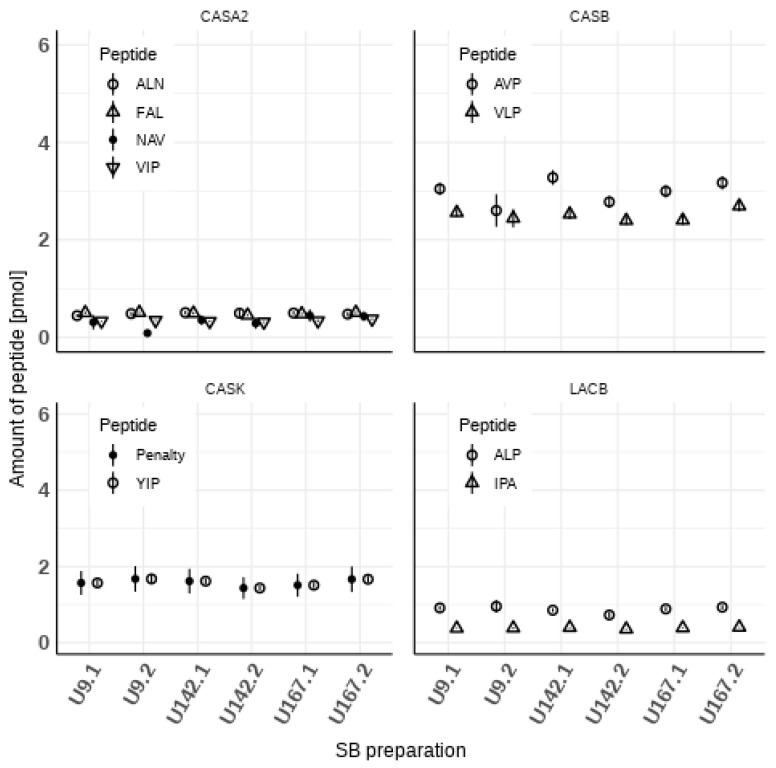
Measured amounts of peptides sorted by the respective protein (CASA2, CASB, CASK, and LACB) in the six SBs (index k = 1 to 6, left to right); symbols depict the quantity values of the respective peptides, the vertical bars the associated expanded uncertainties (might be smaller than the symbol, expansion factor k = 2); for CASK, the penalty applied for the uncertainty of *f_P-P,j,k_* is also displayed.

**Figure 3 foods-11-00869-f003:**
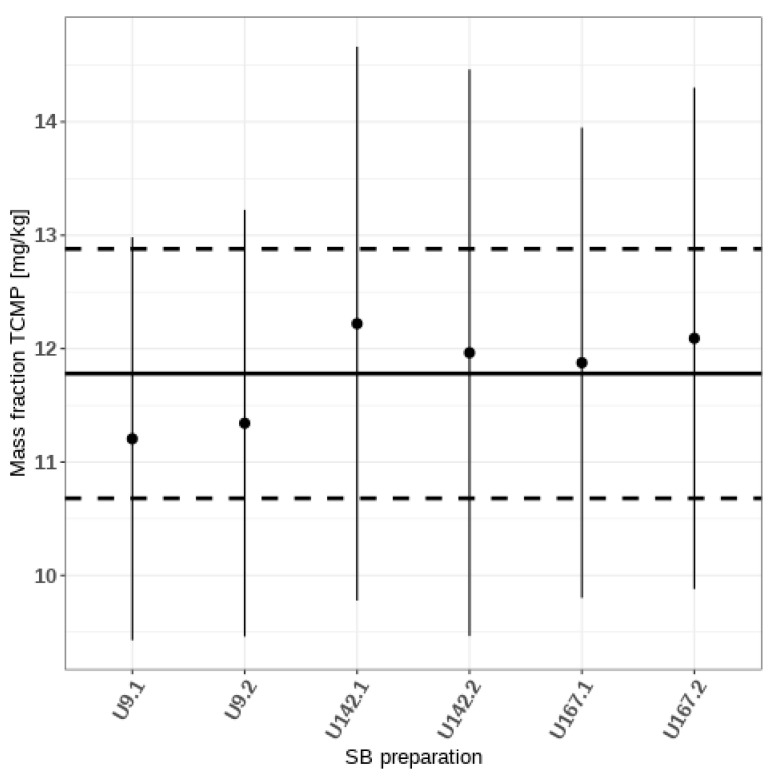
The mass fractions of TCMP in the six SB preparations with associated expanded measurement uncertainties. Solid circles depict the quantity values, and the vertical bars the uncertainties. The bold horizontal line represents the assigned value, and the dashed lines its uncertainty range.

**Table 1 foods-11-00869-t001:** List of the 11 proteotypic peptides of five major constituent proteins of TCMP.

Protein ^a^	Peptide	Position ^b^	*i* ^c^	*j* ^d^
αS1-casein (CASA1_BOVIN)	FFVAPFPEVFGK	38–49	1	1
	YLGYLEQLLR	106–115	2	1
αS2-casein (CASA2_BOVIN)	ALNEINQFYQR	96–106	1	2
	NAVPITPTLNR	130–140	2	2
	FALPQYLK	189–196	3	2
	VIPYVR	215–220	4	2
β-casein (CASB_BOVIN)	VLPVPQK	185–191	1	3
	AVPYPQR	192–198	2	3
κ-casein (CASK_BOVIN)	YIPIQYVLSR	46–55	1	4
β-lactoglobulin (LACB_BOVIN)	IPAVFK	94–99	1	5
	ALPMHIR	158–164	2	5

^a^ protein name (UniProtKB identifier); ^b^ position of the peptide within the amino acid sequence of the protein; ^c^ index *i* of Equations (1) and (2); and ^d^ index *j* of Equations (1)–(3).

**Table 2 foods-11-00869-t002:** Uncertainty budget starting with peptide FFV (*i* = 1) of protein CASA1 (*j* = 1) in SB U167.2 (k = 6).

	Eq.	Term	x	u(x)	Unit	RSU	Index
FFV in TCMP,6	1	*m_Y,_* _6_	2.495 × 10^−1^	1.00 × 10^−4^	g	0.04%	<1%
		*b_Zc_* _2*,*1*,*1_	3.293 × 10^−11^	4.19 × 10^−13^	mol/g	1.3%	92%
		*m_Z_* * _c_ * _2_	2.515 × 10^−1^	1.00 × 10^−4^	g	0.04%	<1%
		*m_Yc_* _2_	2.506 × 10^−1^	1.00 × 10^−4^	g	0.04%	<1%
		*b_Zc_* _1*,*1*,*1_	3.890 × 10^−12^	4.97 × 10^−14^	mol/g	1.3%	1%
		*m_Zc_* _1_	2.495 × 10^−1^	1.00 × 10^−4^	g	0.04%	<1%
		*m_Yc_* _1_	2.512 × 10^−1^	1.00 × 10^−4^	g	0.04%	<1%
		*R′* _1*,*1*,*6_	4.579 × 10^−1^	1.72 × 10^−3^		0.4%	6%
CASA1 in TCMP,6	2	*n* _1*,*1*,*6_	3.334 × 10^−12^	5.00 × 10^−14^	mol	1.5%	<1%
		*n* _2*,*1*,*6_	4.236 × 10^−12^	7.74 × 10^−13^	mol	18%	37%
		*N* _1_	2				
		*m_Exsolv,_* _6_	1.492 × 10^1^	1.30 × 10^−4^	g	0.00%	<1%
		*m_X,_* _6_	1.032	1.30 × 10^−4^	g	0.01%	<1%
		*m_Extr,_* _6_	2.613 × 10^−1^	1.00 × 10^−4^	g	0.04%	<1%
		*M* _1_	2.080 × 10^4^	1.25 × 10^3^	g/mol	6.0%	13%
		*f_P-P,_* _1*,*6_	1	1.19 × 10^−1^		12%	50%
5 proteins in TCMP,6	3	*w* _1*,*6_	4.356 × 10^−6^	7.32 × 10^−7^	g/g	17%	53%
		*w* _2*,*6_	5.925 × 10^−7^	4.88 × 10^−8^	g/g	8.2%	<1%
		*w* _3*,*6_	3.755 × 10^−6^	3.19 × 10^−7^	g/g	8.5%	10%
		*w* _4*,*6_	1.743 × 10^−6^	1.80 × 10^−7^	g/g	10%	3%
		*w* _5*,*6_	6.769 × 10^−7^	2.66 × 10^−7^	g/g	39%	7%
		*f* * _ε_ *	1.087	5.02 × 10^−2^		4.6%	26%
6 independent TCMP values	4	*w_TCMP,_* _6_	1.209 × 10^−5^	1.09 × 10^−6^	g/g	9%	15%
		*w_TCMP,_* _5_	1.187 × 10^−5^	1.04 × 10^−6^	g/g	9%	13%
		*w_TCMP,_* _4_	1.196 × 10^−5^	1.25 × 10^−6^	g/g	10%	19%
		*w_TCMP,_* _3_	1.222 × 10^−5^	1.22 × 10^−6^	g/g	10%	19%
		*w_TCMP,_* _2_	1.134 × 10^−5^	9.41 × 10^−7^	g/g	8%	11%
		*w_TCMP,_* _1_	1.120 × 10^−5^	8.88 × 10^−7^	g/g	8%	10%
		*f_SB-SB_*	1	1.43 × 10^−2^		1.4%	13%
Result		*x_a_*	11.8	0.55	mg/kg	4.7%	

x—quantity value of respective term of respective Eq., u(x)—standard uncertainty associated with x, RSU—relative standard uncertainty, and index—uncertainty contribution index [22]. The first row of each subsequent block states the combined uncertainty *u_c_* of the previous block; the row “results” states the *u_c_* of all contributors incl. *u_hom_* (Equation (6)). For the meaning of the indexes *i*, *j*, and *k,* see Table 1.

## Data Availability

Not applicable.

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
