# Peer review of "Assignment of a Reference Value of Total Cow’s Milk Protein Content in Baked Cookies Used in an Interlaboratory Comparison"

_foods, 2022, doi:10.3390/foods11060869_

Round 1

Reviewer 1 Report

This manuscript reports interlaboratory comparison in determination of total cow’s milk protein in baked cookies. Presented results are not novel, similar approach authors presented in previous paper titled “Total cow’s milk protein in cookies: the first interlaboratory comparison with a well-defined measurand fit for allergen risk assessment”.

  • Figure 1 – What is the cause differences between uncertainties between FFV and YLG ?
  • Page 10 line 354 – authors shows moisture loss during the baking, but they not explain how it was determined and what was uncertainty of those measurements.

Author Response

This manuscript does not report the results of an Interlaboratory Comparison (ILC) but the determination of the assigned value for the test material that was used in that ILC. “Total cow’s milk protein in cookies: the first interlaboratory comparison with a well-defined measurand fit for allergen risk assessment” is a paper reporting on the outcome. As such this manuscript is not a duplication but a necessary addition.

The cause of the differences between uncertainties between FFV and YLG is explained on lines 103ff and 292. In brief, the uncertainty of the purity assessment for YLG was estimated at 16 % while for FFV it was below 2.5 % and this difference causes the error bars in Fig. 1 to be so different.

The moisture loss was determined by weighing cookies before and after backing. No details were provided because it has no bearing on the reported value other than providing some sort of verification.

Reviewer 2 Report

This is a well conducted and performed work dealing with the assessment of a reference value for milk proteins (caseins + lactoglobulin) in cookies, for interlaboratory comparisons within EU. The methodology is nicely described and results issued from LC-MS measurements using labelled standard peptides are well analyzed and discussed.

I have only minor comments:

  • Why peptide(s) derived from lactalbumin, another allergen from milk, were not used?
  • What is known about the milk allergen threshold values susceptible to trigger some allergic reaction in sensitized people?
  • Does the food product matrix has any influence on the LC-MS measurements?

Author Response

Lactalbumin only represents about 2 % of the total milk protein. Adding it to the assay would have not had a large added value but cost a lot of additional effort. Therefore, we did not add lactalbumin peptides.

Threshold values triggering an adverse reaction in sensitive humans were not subject of this manuscript and are therefore not dealt with. Having said that there is the VITAL program which provides information about possible threshold values (https://vital.allergenbureau.net/).

The food matrix has a significant influence on LC-MS measurements which is why IDMS was employed.

Reviewer 3 Report

Manuscript is straighforward methods development. I would remind authors that the first occurance of abbreviation needs to written out completely within the text (not abstract).

Author Response

In attempting to build the uncertainty budget we described in detail different steps of the method of analysis.  This might create the impression of a method development description but it is not.

The abbreviations in the main text have all been defined before first use. We consider ELISA to be a common enough abbreviation that it does not need a definition.